# Aerosol characterization in the Subtropical Eastern North Atlantic region derived from long-term AERONET measurements

Africa Barreto[1,2], Rosa D. García[3,1], Carmen Guirado-Fuentes[2,1,a], Emilio Cuevas[1], A. Fernando Almansa[4,1], Celia Milford[1], Carlos Toledano[2], Francisco J. Expósito[5], Juan P. Díaz[5], and Sergio F. León-Luis[3,1]

[1]Izaña Atmospheric Research Center (IARC), Agencia Estatal de Meteorología (AEMET), Spain
[2]Atmospheric Optics Group of Valladolid University (GOA–UVa), Valladolid University, Valladolid, Spain
[3]TRAGSATEC, Madrid, Spain
[4]Cimel Electronique, Paris, France
[5]Departamento de Física, Universidad de La Laguna (ULL), Canary Islands, Spain
[a]Now at: Servicio de Evaluación del Servicio Canario de la Salud, Canary Islands, Spain

**Correspondence:** Africa Barreto (abarretov@aemet.es)

**Abstract.** A comprehensive characterization of atmospheric aerosols in the Subtropical Eastern North Atlantic has been carried out using long-term ground-based Aerosol Robotic NETwork (AERONET) photometric observations from a unique network made up of four stations strategically located from the sea level to 3555 m height on the island of Tenerife over the period 2005-2020. This site can be considered a sentinel of the passage of airmass going to Europe from Africa and therefore the

aerosol characterization performed here adds important information to analyse their evolution during the path toward Northern Europe. Two of these stations (Santa Cruz de Tenerife –SCO- at sea level and La Laguna –LLO- at 580 m asl) are located within the Marine Atmospheric Boundary Layer (MABL) and the other two (Izaña –IZO- at 2373 m asl and Teide Peak –TPO- at 3555 m asl) are high mountain stations within the Free Troposphere (FT). Monthly climatology of Aerosol Optical Depth (AOD), Ångström Exponent (AE), aerosol concentration, size distribution, and aerosol optical properties has been obtained

for the MABL and FT. Quite consistent measurements at the four sites have been used to categorise the main atmospheric scenarios confirming the predominance of the alternating background to dust-loaded Saharan air mass conditions seasonally affecting the sites as a result of the seasonal dust transport over the Subtropical North Atlantic. Background conditions prevail in the MABL and FT most of the year, while dust-laden conditions dominate in July and August.

The MABL under background conditions appears as a well-mixed layer with low aerosol concentration (volume concentra-

tion, $VolCon$, ranging from 0.02±0.01 to 0.04±0.02 $\mu m^3 \cdot \mu m^{-2}$) with a predominance of coarse mode marine aerosols (effective radius, $Reff$, changing from 1.60±0.19 to 1.91±0.34 $\mu m$) and volume contribution of the fine mode fraction $Vf/Vt <$ 0.35. The clean FT has been characterised by remarkably low aerosol loading and a predominant impact of fine mode aerosols throughout the year ($Vf/Vt$ with a maximum value of 0.93±0.13) with an average $Reff$ of 0.16±0.02 $\mu m$. However, under dust-laden conditions, we observe the predominance of coarse mode aerosols, mainly in summer, with maximum $VolCon$

values of 0.26±0.23 $\mu m^3 \cdot \mu m^{-2}$ for MABL and 0.16±0.12 (0.06±0.05) $\mu m^3 \cdot \mu m^{-2}$ for IZO (TPO) and a similar and quite consistent fine mode fraction of 0.12±0.03 in the vertical within MABL and FT. Similarities in micro-physical and optical intensive aerosol properties confirm the Saharan Air Layer (SAL) as a well-mixed layer in terms of the particulate composi-

tion. An estimation of the difference in the aerosol loading in the 1-km layer between IZO and TPO (in terms of $VolCon$ and AOD) is performed in this study, showing that aerosol loading at IZO is double that of TPO but with similar fine mode fraction, effective radius and optical intensive properties. The long-term trend analysis at SCO shows a negative significant trend in the fine AOD mode between 2005 and 2020 (-1.8$\pm$0.5) $\cdot 10^{-5}$ yr$^{-1}$, which might be linked to the large reduction of oil refining $SO_2$ emissions at SCO refinery in 2012.

## 1 Introduction

Tropospheric aerosols impact climate by direct scattering and absorption of the incoming solar radiation and by an indirect effect related to their impact on cloud microphysics. The most recent estimates of the radiative forcing exerted by anthropogenic aerosols on climate confirm the assessment that it is virtually certain that the total aerosol effective radiative forcing (ERF) is negative (Arias et al., 2021). This cooling effect of atmospheric aerosols, set at –1.3 [–2.0 to –0.6] $Wm^{-2}$, partly counteracts the warming effects of anthropogenic greenhouse gases, and is considered as the largest uncertainty in the effect of short-lived climate forcers in future climate projections (Arias et al., 2021).

Aerosol observations from surface networks and satellite-based systems have been enhanced and expanded considerably over the last decades, contributing to the improvements in the understanding and quantification of the net effect of aerosols on climate. Different approaches with diverse temporal and spatial scales are complementary pieces of knowledge to overcome the main difficulties in the study of the aerosol ERF: their highly variable concentration, composition and distribution over space (horizontally and vertically) and time (Toledano et al., 2007; Putaud et al., 2010; Laj et al., 2020). In this regard, AERONET (AErosol RObotic Network, Holben et al. (1998); Giles et al. (2019)) is the major ground-based aerosol network providing globally distributed and near-real-time aerosol observations, freely available for the scientific community. The aerosol optical depth (AOD) is a key variable to study the aerosol radiative forcing. However, due to the lack of scattering information contained in the AOD observations, more information extracted from the angular distribution of sky radiances is important to properly understand the aerosol radiative effect (Dubovik and King, 2000; Kok et al., 2017; Torres et al., 2017). This information is critical to validate aerosol models as well as to assess the prescribed attributes usually given to atmospheric aerosols in current models or inversion schemes (Dubovik et al., 2006; Kok et al., 2017; Torres et al., 2017). There are other surface networks such as SKYNET (Sky Radiometer Network, Takamura and Nakajima (2004); Nakajima et al. (2020)) or GAW-PFR (Global Atmospheric Watch-Precision Filter Radiometer Network, Wehrli (2000)) that, although less extensive, are also capable of providing very useful information for aerosol monitoring.

In this study, we describe the long-term seasonal evolution of atmospheric aerosols by using AERONET observations at four different sites, at different altitudes in Tenerife, in the Subtropical Eastern North Atlantic region. This region can be considered a key location for aerosol monitoring because it is in the path of long-range transports such as mineral dust from Sahel-Sahara regions (Carlson and Prospero, 1972; Prospero and Carlson, 1972; Tsamalis et al., 2013; Cuevas et al., 2015; Rodríguez et al., 2015; Rodríguez et al., 2020; Barreto et al., 2022), dust from North America (García et al., 2017), or sulfates, biomass burning and other pollutants from North America, Europe or Africa (Viana et al., 2002; Basart et al., 2009; Rodríguez et al., 2011;

García et al., 2017; Rodríguez et al., 2020; Wang et al., 2021). On the northern edge of the dust belt in summer but still affected by dust transport in winter (Alonso-Pérez et al., 2007, 2011, 2012; Rodríguez et al., 2011; Cuevas et al., 2015), this region presents a stronger seasonal dependence in dust transport than tropical latitudes, representative of the almost pure Saharan dust in summer and winter (Barreto et al., 2022). Furthermore, the strong vertical stratification in the lower troposphere typical of this eastern side of the Subtropical North Atlantic implies the presence of several layers and transition levels with different vertical humidity and temperature gradients, strongly affecting the aerosol layering (Carrillo et al., 2016; Barreto et al., 2022). A humid and relatively cold Marine Atmospheric Boundary Layer (MABL) is well-differentiated in the lowermost troposphere, limited on the top by a strong temperature inversion layer, with a dry and relatively warm free troposphere (FT) above (Font-Tullot, 1956; Cuevas, 1995; Carrillo et al., 2016, and references therein). Most of the year, the Trade Wind Layer (TWL) separates the MABL and the FT as a consequence of the quasi-permanent subsidence conditions modulated by the descending branch of the Hadley cell (Carrillo et al., 2016). The contrasting aerosols regimes observed at this site and the very stable and low aerosol turbidity within the FT make it an excellent site for aerosol monitoring and calibration (Toledano et al., 2018; Cuevas et al., 2019b). Not in vain, Izaña Observatory, located in the FT, is considered one of the two absolute calibration sites in the world for both AERONET and GAW-PFR global networks (Toledano et al., 2018; Cuevas et al., 2019b).

The main objective of this paper is to perform a characterization of atmospheric aerosols in terms of optical and micro-physical properties using long-term records (2005-2020) in two stations within the MABL (Santa Cruz de Tenerife -SCO- and La Laguna -LLO-) and an additional two stations more within the FT (Izaña -IZO- and Teide Peak -TPO-). These four databases provide high-quality information on aerosols over a period between 9 and 16 years. The possible variation of aerosol properties with height and the different seasonality of aerosols as a consequence of the main aerosol transports over this region are also studied. In this regard, background and dust-laden conditions have been identified and characterized as the predominant conditions in the four sites. Sect. 2.1 and 2.2 describe the experimental sites, aerosol data sets and instrumentation used in this work. The main results are shown in Sect 3. Sect 3.1 is dedicated to the seasonal characterization of optical aerosol properties in the MABL and FT in terms of AOD and Angström Exponent (AE). Sect 3.2 describes the seasonal characterization of aerosol optical and micro-physical properties in these two atmospheric layers from photometric inversion products. In Sect 3.3, a preliminary trend analysis of key optical micro-physical properties is evaluated for Santa Cruz and Izaña Observatories. These two stations have been selected due to their long and high-quality aerosol databases, representative of MABL and FT conditions, respectively. Finally, the main conclusions of this study are summarized in Sect. 4.

## 2 Sites and instrumentation

### 2.1 The Sites

The island of Tenerife is located in the Subtropical Eastern North Atlantic region (Fig. 1 (a)) under the influence of the northeast trade wind regime, which causes two well-differentiated layers to be present in the lowermost Subtropical North Atlantic troposphere. A humid and relatively cold MABL limited on its top by a strong temperature inversion layer is capped by a very

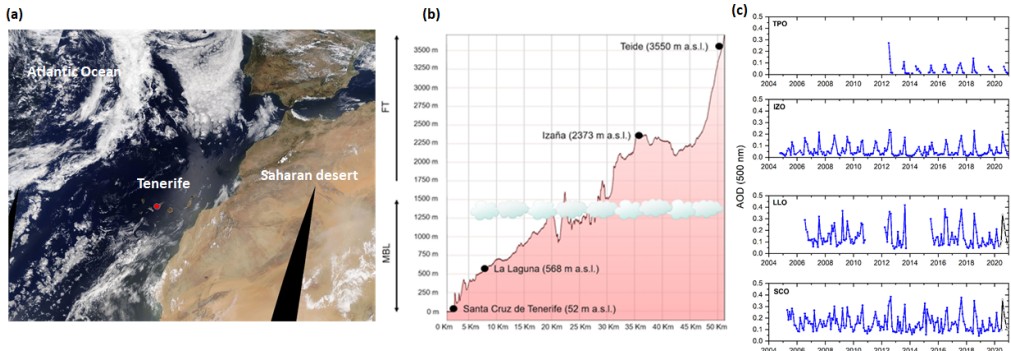

**Figure 1.** (a) Modis visible imagery on 26 August 2021 over western Africa and the northern Atlantic ocean. The red dot indicates the location of the Tenerife island. Image credits NASA Worlwide (https://worldview.earthdata.nasa.gov). (b) The elevation profile of Tenerife Island, indicating the corresponding elevation of the four stations (SCO, LLO, IZO and TPO) and their location regarding Marine Atmospheric Boundary Layer (MABL) and Free Troposphere (FT). Stratocumulus cloud top is limited by the trade wind inversion. (c) Time series of monthly mean AOD at 500 nm at SCO, LLO, IZO and TPO. The blue dots represent the AERONET version 3.0 level 2.0 AOD data and black dots the level 1.5.

dry FT above. SCO and LLO are located in the MABL, while IZO and TPO are located in the FT, normally above a temperature inversion layer.

90 Ground-based aerosol observations from four AERONET stations located at different altitudes on Tenerife (Canary Islands, Spain) have been used in this work (Fig. 1). These stations (Fig. 1 (b)), located with a maximum horizontal distance between them of 50 km, are the following:

- Santa Cruz de Tenerife Observatory (SCO; 28.5°N, 16.2°W, 52 m a.s.l.) is a coastal urban station (Cuevas et al., 2019a) located in the center of Santa Cruz de Tenerife and very close to the city harbour. Following Basart et al. (2009), marine coarse aerosols are predominant at this site throughout the year, while Saharan dust contribution is predominant from winter to spring due to the frequent dust outbreaks over this region. However, the portion of fine mode aerosols from local (urban or industrial) activities is lower than expected for such an urban station because of the dispersion of pollutants by the predominant trade-wind regime and the sea breeze circulation during daylight (Rodríguez et al., 2008).

- La Laguna Observatory (LLO; 28.5°N, 16.3°W, 568 m a.s.l.) is an urban station far away from industrial activities. North-westerly winds are the prevailing regime, leading to a cloudy and wet climate except under the influence of Saharan air masses when humid north-easterly air masses are displaced by dryer ones from the African continent.

- Izaña Observatory (IZO; 28.3°N, 16.5°W, 2373 m a.s.l.) is located on a mountain plateau with no significant local pollution sources. It is normally above the temperature inversion layer and dominated by north-westerly winds with a very dry and stable atmosphere with clear sky and clean air (pristine) conditions. It is affected by mineral dust when

the Saharan Air Layer (SAL) top exceeds the station height, mainly in summer. Despite the latter, it is an excellent site for remote sensing atmospheric research and monitoring. IZO enrolled in the World Meteorological Organization (WMO) Global Atmosphere Watch (GAW) programme in 1989, and it has contributed to several international networks such as GAW-PFR since 2001, and AERONET since 2004, as one of the two absolute AERONET calibration sites (https://aerospain.aemet.es/, last access: 22 March 2022). In July 2014, IZO was appointed a WMO - Commission for Instruments and Methods of Observations Testbed for Aerosols and Water Vapor Remote Sensing Instruments (WMO-CIMO, WMO (2014)). More details, of the measurement programmes can be found in Cuevas et al. (2019a).

– Teide Peak Observatory (TPO; 28.3°N, 16.6°W, 3550 m a.s.l.) is located at the cable car terminal Teide volcano in the Teide National Park. TPO is characterized by extremely pristine conditions and, similarly to IZO, is affected by mineral dust when the SAL top exceeds the TPO height, mainly in summer. TPO was established as a satellite station of IZO in 2012 (Cuevas et al., 2019a).

SCO, IZO and TPO stations are managed by the Izaña Atmospheric Research Centre (IARC), which is part of the State Meteorological Agency of Spain (AEMET; more information at http://izana.aemet.es; last access: 22 March 2022), while LLO is managed by La Laguna University (https://www.ull.es; last access: 22 March 2022). SCO, LLO and IZO are devoted to continuous long-term monitoring. AERONET measurements at TPO, due to adverse weather conditions, are mainly available between mid-spring and mid-autumn, having continuous records from September 2020.

## 2.2 Cimel sun photometer data sets

In this study, aerosol measurements were obtained from two different Cimel sun photometer CE318 versions: CE318-N (Holben et al., 1998) and CE318-TS (Barreto et al., 2016; Giles et al., 2019). Ground-based CE318 sun measurements were performed at eight or nine nominal wavelengths (340 to 1640 nm) with an approximate field of view of $\sim 1.3°$(Holben et al., 1998; Torres et al., 2013) and 10 nm full-width-at-half-maximum (FWHM) bandwidth, except for 340 and 380 nm which have 2 and 4 nm FWHM, respectively, and for 1640 nm, with an FWHM of 25 nm. AOD and AE have been retrieved as products from direct measurements. The AOD total uncertainty is approximately 0.01-0.02 for field sun photometers and 0.002-0.009 for reference instruments (both spectrally dependent, with higher errors in the UV) (Eck et al., 1999; Holben et al., 2001; Giles et al., 2019). AE represents the AOD spectral dependence and is a qualitative indicator of the predominant aerosol size (Ångström, 1929; Eck et al., 1999). Linear fit determination of AE in the range 440–870 nm (440 nm, 500 nm when available, 670 nm, and 870 nm) has been used ($AE_{440-870nm}$).

Aerosol micro-physical and optical properties obtained from the AERONET inversion algorithm are also analyzed: particle volume size distribution, volume particle concentration ($VolCon$), fine mode volume fraction ($V_f/V_t$), effective radius ($R_{eff}$), single scattering albedo ($SSA$), refractive index, and asymmetry parameter ($g$). Dubovik and King (2000), Dubovik et al. (2006) and Sinyuk et al. (2020) describe AERONET retrieval, measurement accuracy, and error estimates. It should be noted that AERONET level 2.0 retrievals for $SSA$ and imaginary refractive index are limited to $AOD_{440nm} > 0.4$ and solar zenith angles $> 50°$, which limits strongly the amount of data available for aerosol characterization (Sinyuk et al., 2020).

AERONET version 3.0 level 2.0 (or level 1.5 depending on data available at the station) dataset (Sinyuk et al. (2020); https://aeronet.gsfc.nasa.gov, last access: 22 March 2022) has been used in this work (Fig. 1). More specifically, AERONET
level 2.0 data series at IZO is available from October 2004 to December 2020. This data set is composed of records from a total of 16 reference instruments in the period 2004-2020. However, the homogeneity and quality of this AERONET-Cimel AOD data series at Izaña has been confirmed by Cuevas et al. (2019b) using a long-term AOD and AE comparison with the three GAW-PFR reference instruments, WMO AOD reference, running at Izaña at the same time period, and also by Toledano et al. (2018), who assessed the suitability of Izaña as a Langley plot calibration site using 15 years of Langley calibrations.
AERONET level 2.0 measurements at SCO and LLO are available from April 2005 and July 2006, respectively, until June 2020. Level 1.5 is used from June to December 2020. Both datasets are obtained from field instruments that are replaced every year. Regarding TPO, the data series is composed of discontinuous records from field instruments since July 2012. Level 2.0 is available until December 2020.

## 3 Results

### 3.1 MABL and FT AOD and AE aerosol seasonal characterization

Monthly mean AOD at 440, 500, 675, 870 and 1020 nm and $AE_{440-870nm}$ are shown in Fig. 2 for the four sites in Tenerife. SCO (Fig. 2 (a)) and LLO (Fig. 2 (c)) display low $AOD_{500nm}$ values in May (0.12±0.09 at SCO and 0.09±0.07 at LLO) and between October and February, with values of 0.12±0.03 at SCO and 0.10±0.03 at LLO. In those months, mean $AOD_{500nm}$<0.15 and $AE_{440-870nm}$ values are between 0.5 and 0.75, indicating an atmosphere dominated by marine aerosol (Fig. 2 (b) and 2 (d))
(e.g., Holben et al. (2001); Dubovik et al. (2002); Smirnov et al. (2002); Basart et al. (2009)). Higher $AOD_{500nm}$ values are recorded in July (0.24±0.22 at SCO and 0.26±0.23 at LLO) and August (0.26±0.23 at SCO and 0.25±0.25 at LLO) together with mean $AE_{440-870nm}$<0.5 for both stations, due to the presence of the SAL over Tenerife during these months (Barreto et al., 2022). It is important to note that the maximum AOD standard deviation found in the summer months (June to August) is a consequence of the strong AOD variations due to the seasonal dust transport (Prospero and Carlson, 1980; Prospero, 1996;
Karyampudi et al., 1999; Engelstaedter et al., 2006). A secondary maximum of $AOD_{500nm}$ (0.16±0.16 at SCO and 0.15±0.17 at LLO) and a minimum of $AE_{440-870nm}$ (0.54±0.30 at SCO and 0.62±0.35 at LLO) is observed in March when mineral dust is transported at lower altitudes directly affecting the MABL (Barreto et al., 2022). Our results show a MABL characterised by a marked seasonality due to mineral dust transport at these latitudes in summer with a predominant influence of marine aerosols the rest of the year. Consistent results between the two stations located at different altitudes within the MABL indicate
this is a well-mixed layer with similar aerosol loading and particle sizing.

Regarding the FT, low mean $AOD_{500nm}$ values are observed between October and February (average values of 0.03±0.04) at IZO (Fig. 2 (e)) and between October and December (0.02±0.01) at TPO (Fig. 2 (g)). Accordingly, $AE_{440-870nm}$ values of 1.01±0.34 at IZO (Fig. 2 (f)) and 1.21±0.25 at TPO (Fig. 2 (g)) are also observed, indicating an extremely clean atmosphere with predominant fine aerosols. In contrast to these dominating background conditions, higher $AOD_{500nm}$ and lower
$AE_{440-870nm}$ values are recorded in July ($AOD_{500nm}$ of 0.15±0.16 and $AE_{440-870nm}$ of 0.54±0.47 at IZO; 0.10±0.14 and

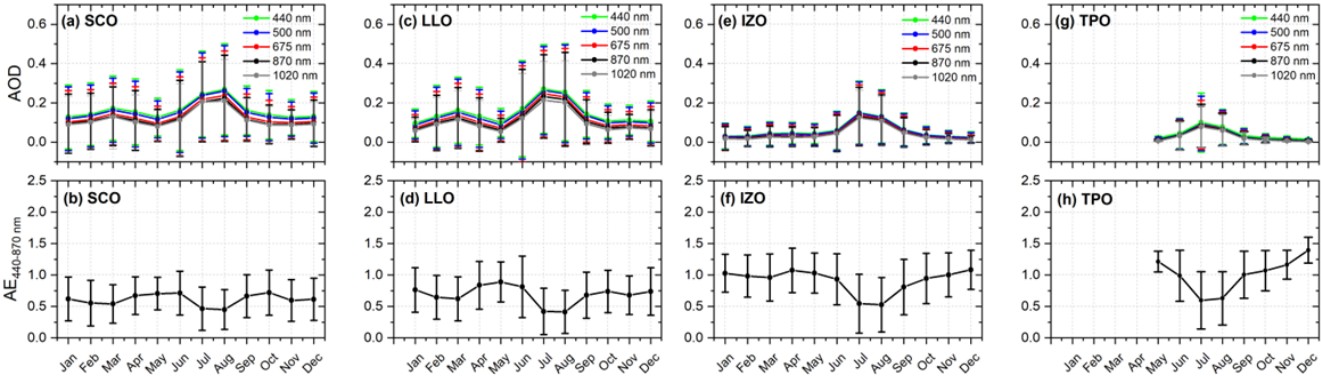

**Figure 2.** Monthly mean aerosol optical depth (AOD) at 440, 500, 675, 870 and 1020 nm and Ångström Exponent ($AE_{440-870nm}$) SCO from April 2005 and December 2020 ((a) and (b)), at LLO between July 2006 and December 2020 ((c) and (d)), at IZO between from October 2004 and December 2020 ((e) and (f)) and at TPO between July 2012 and December 2020 ((g) and (h)). Error bars indicate the standard deviation.

0.60±0.45 at TPO) and August (0.13±0.14 and 0.53±0.43 at IZO; 0.07±0.09 and 0.63±0.42 at TPO). Those records indicate the presence of larger particles, when mineral dust is transported over Tenerife at a high altitude within the SAL at this time of the year (Carlson, 2016; Barreto et al., 2022).

Our results, as well as those published by other authors (Rodríguez et al., 2011; Cuevas et al., 2019b; Barreto et al., 2022),
confirm the predominance of the alternating situations from background conditions characterised by fine aerosols and remarkably stable AODs to dust-laden conditions with coarse mode particles at the four sites. We have therefore used the criterion based on $AOD_{500nm}$ and $AE_{440-870nm}$ thresholds defined by Barreto et al. (2022) to perform a more detailed classification of the predominant atmospheric scenarios at these sites.

The data set corresponding to $AOD_{500nm}<0.15$ and $AE_{440-870nm}>0.50$ at SCO has been selected for the study of back-
ground conditions in the MABL (41% for SCO and 50% for LLO of the total measurements) (Barreto et al., 2022). $AOD_{500nm} \geq 0.15$ and $AE_{440-870nm} \leq 0.50$ at SCO has been selected for the study of dust-laden conditions in the MABL (24% and 26% for SCO and LLO, respectively) (Fig. 3 (a) and 3 (b)). The rest of the cases have been classified as "mixed aerosols". Our results indicate that background conditions prevail in the MABL most of the year (Fig. 4 (a)), particularly in May and June (more than 15 days per month) as previously reported by Cuevas et al. (2015), who identified clean atmosphere and sporadic dust intrusions during
these two months by the analysis of lidar vertical profiles at SCO. In July and August, dust-laden conditions dominate while between October and May dust-laden air masses are less frequent with the exception of a slight increase in March.

Regarding the FT, background conditions are identified as those at IZO with $AOD_{500nm} \leq 0.10$ and $AE_{440-870nm} \geq 0.60$ as threshold values (Barreto et al., 2022).

This data selection comprises $\sim73\%$ of total observations (Fig. 3 (c) and 3 (d)). For the identification of dust-laden conditions
we have set $AOD_{500nm} \geq 0.10$ and $AE_{440-870nm} \leq 0.60$ as threshold values.

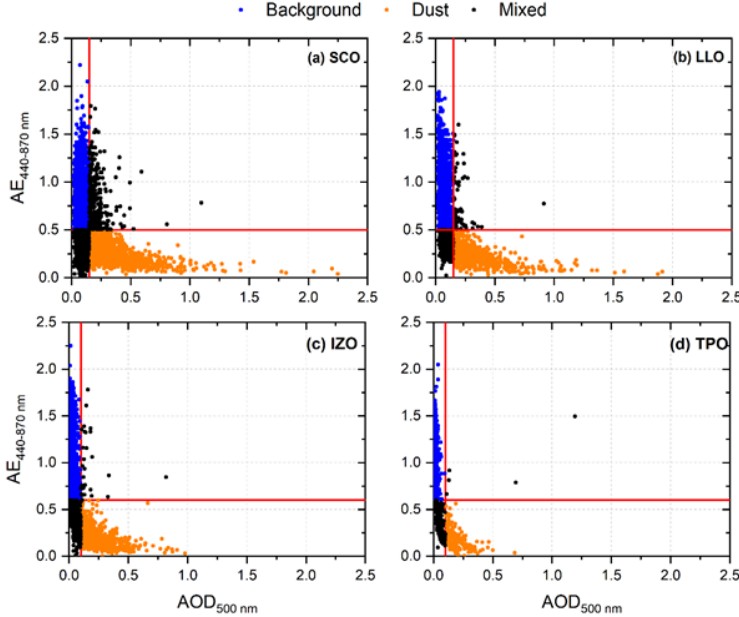

**Figure 3.** Scatterplot of the daily values of $AE_{440-870nm}$ versus $AOD_{500nm}$ at (a) SCO, (b) LLO, (c) IZO and (d) TPO. The red lines indicate the threshold limits established for background conditions at SCO and LLO: $AOD_{500nm}$ <0.15 and $AE_{440-870nm}$>0.50 (blue dots) and for the dust-laden scenario: $AOD_{500nm} \geq 0.15$ and $AE_{440-870nm} \leq 0.50$ (orange dots). Similarly, in (c) and (d) the red lines indicate the threshold limits and IZO and TPO: $AOD_{500nm}$ <0.10 and $AE_{440-870nm}$>0.60 (blue dots) and dust-laden: $AOD_{500nm} \geq 0.10$ and $AE_{440-870nm} \leq 0.60$ (orange dots). Black dots indicate the presence of mixed aerosols.

Dust-laden conditions, as defined in this paper, follow a seasonal pattern displayed in Fig. 4 (b). IZO is under background FT conditions mostly more than 50% of days every month, except in July and August when the number of days under dust-laden and background conditions are quite similar. The highest frequency of background conditions is found in April, May and June (21, 25 and 22 days per month, respectively). Dust conditions are scarce from October to February when dust transport hardly reaches the altitude of the station (Fig. 4 (b)).

## 3.2 Extensive MABL and FT aerosol characterization based on photometric inversion products

Optical and micro-physical aerosol properties inferred from AERONET inversion products have been used in this study to incorporate some important information for climate and radiative studies (Dubovik and King, 2000; Dubovik et al., 2006; Boucher et al., 2013). Monthly mean aerosol volume size distributions ($dV/dlnR$) for background and dust-laden conditions are displayed in Figs. 5 and 7, while monthly mean aerosol volume concentration ($VolCon$) and effective radius ($Reff$) for total, fine and coarse modes in addition to fine mode fraction ($Vf/Vt$), are presented in Figs. 6 and 8, respectively. Fig. 9 also represents other inversion products at the four stations but only for dust-laden (high aerosol loading) conditions: asymmetry factor ($g$), single scattering albedo ($SSA$) and the real ($n$) and imaginary ($k$) parts of the refractive index at 440, 675, 870

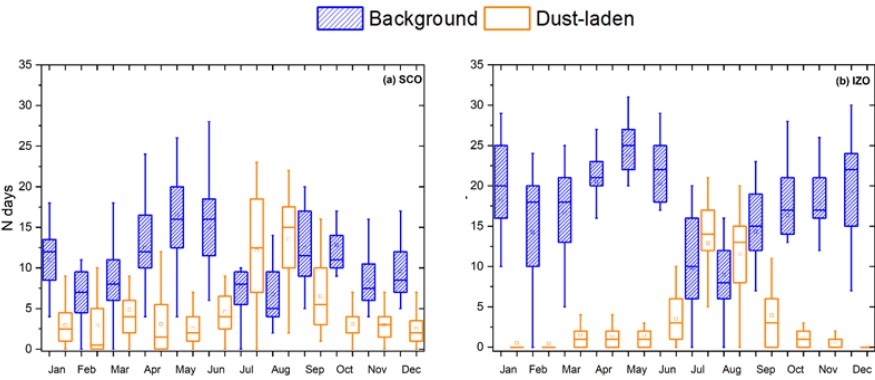

**Figure 4.** Number of days per month under background and dust-laden conditions at (a) SCO and (b) IZO observatory. Lower and upper boundaries for each box are the $25^{th}$ and $75^{th}$ percentiles, the solid line is the median value, the point represents the mean value and the crosses indicate values out of the 1.5-fold box area (outliers). Hyphens are the maximum and minimum values.

and 1020 nm. The reason for not including these parameters for background conditions is the high uncertainty in AERONET inversion products under low aerosol loading as reported by Sinyuk et al. (2020).

### 3.2.1 Background conditions

The MABL is characterised by a bimodal lognormal size distribution with a coarse mode predominant throughout the year, slightly increased in the summer and spring months in the case of LLO (Fig. 5a and 5b). These characteristics are quite consistent between the two stations within the MABL, confirming effective vertical mixing within the cloud-free subtropical MABL (Carrillo et al., 2016; Barreto et al., 2022). This predominant coarse mode is also evident from the low fine fraction ($Vf/Vt$) observed at the two stations in Fig. 6 (c) and (f) (below 0.35 throughout the year). These features are in agreement with the results presented by Smirnov et al. (2002) and Dubovik et al. (2002) for a marine background environment. Relatively stable and low total $VolCon$ values are observed during the year, between $0.02\pm0.01\mu m^3\cdot\mu m^{-2}$ and $0.04\pm0.02\ \mu m^3\cdot\mu m^{-2}$ in the two stations. A quite consistent analysis in terms of $Reff$ between these two stations is also shown in Figs. 6 (b) and (e) for SCO and LLO, respectively. Fine mode aerosols with a $Reff$ of $0.15\pm0.02\ \mu m$ seem to be present at the two stations throughout the year, with a $Reff$ of coarse aerosols with small seasonal dependence: minimum values ($1.60\pm0.19\mu m$) in late spring-early summer and maximum values ($1.91\pm0.34\ \mu m$) in winter. Average fine-mode $Reff$s are $0.15\pm0.02\ \mu m$ ($0.15\pm0.02\ \mu m$) for SCO (LLO) and $1.73\pm0.23\ \mu m$ ($1.75\pm0.27\ \mu m$) for SCO (LLO) in the case of coarse-mode $Reff$s. These values are in agreement with the $Reff$ values reported by Smirnov et al. (2002) and Sayer et al. (2012) at Atlantic sites ($0.12$-$0.16\ \mu m$ for fine mode and $1.69$-$1.93\ \mu m$ for coarse aerosols).

Regarding the FT, Figs. 5 (c) and (d) shows background conditions with remarkably low aerosol loading characterised by a slight bimodality of the aerosol particle size distribution, seasonal dependent. A dominant fine mode is present throughout the

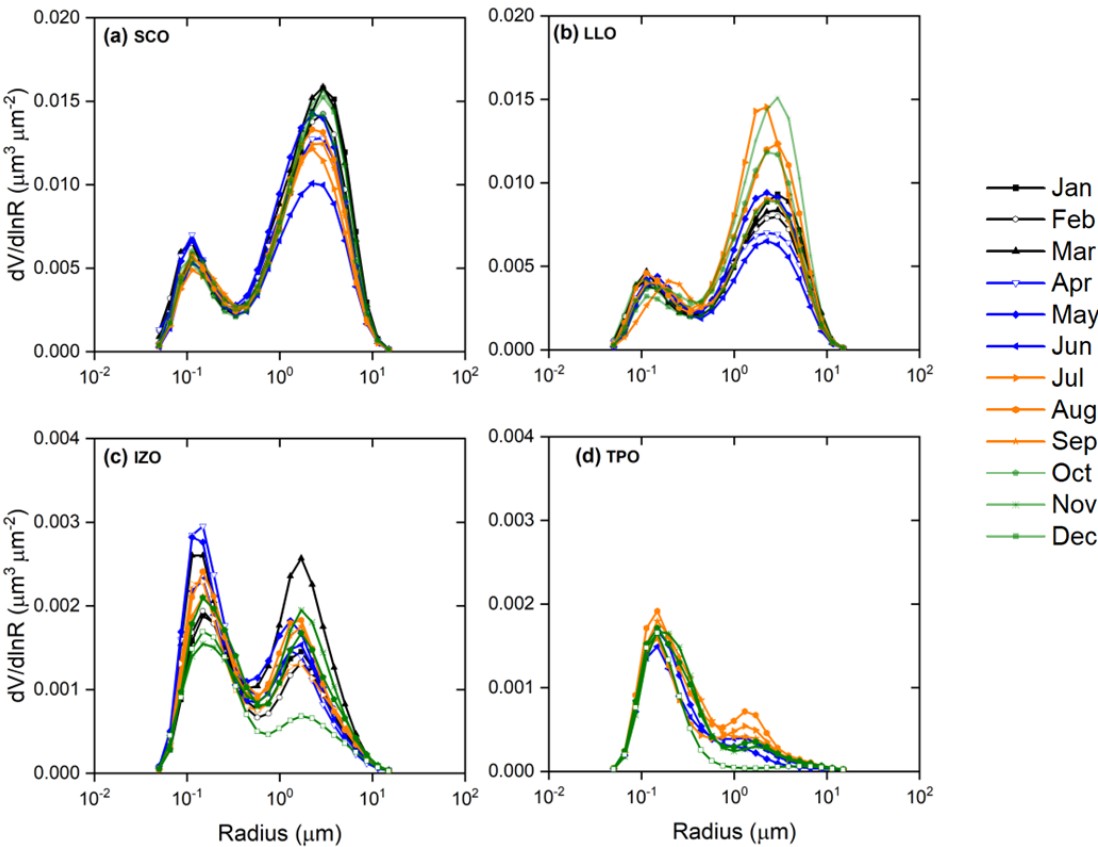

**Figure 5.** Monthly mean aerosol particle size distribution at a) SCO, b) LLO, c) IZO and d) TPO under background conditions.

year. These results are consistent with the high $AE_{440-870nm}$ values observed for these two sites in Sect. 3.1 and the $VolCon$ values in Figs. 6 (g) and (j) for IZO and TPO, respectively.

The dominance of fine mode aerosols (average $Vf/Vt$ of 0.72±0.25 in IZO and 0.84±0.16 in TPO, with a maximum value of 0.93±0.13 in November in TPO) with average fine $Reff$ of 0.16 ±0.02$\mu m$ is observed in Figs. 6 (h) and (i) for IZO, and (k) and (l) for TPO. Note that mean values at TPO correspond to a fraction of the year, from May to December.

The presence of some residual dust on those days considered under background conditions can play a role in aerosol characterization, especially when desert dust intrusions are more frequent. The presence of recirculated dust (with lower $Reff$)

might be the reason for the small decrease in aerosol radius observed in the MABL and FT during late spring-summer months.

### 3.2.2    Dust-laden conditions

The seasonal evolution of the volume size distribution and the most important inversion products in the MABL (SCO and LLO) under dust-laden conditions are displayed in Figs. 7, 8 and 9. Significant seasonal changes are observed in the two stations. A

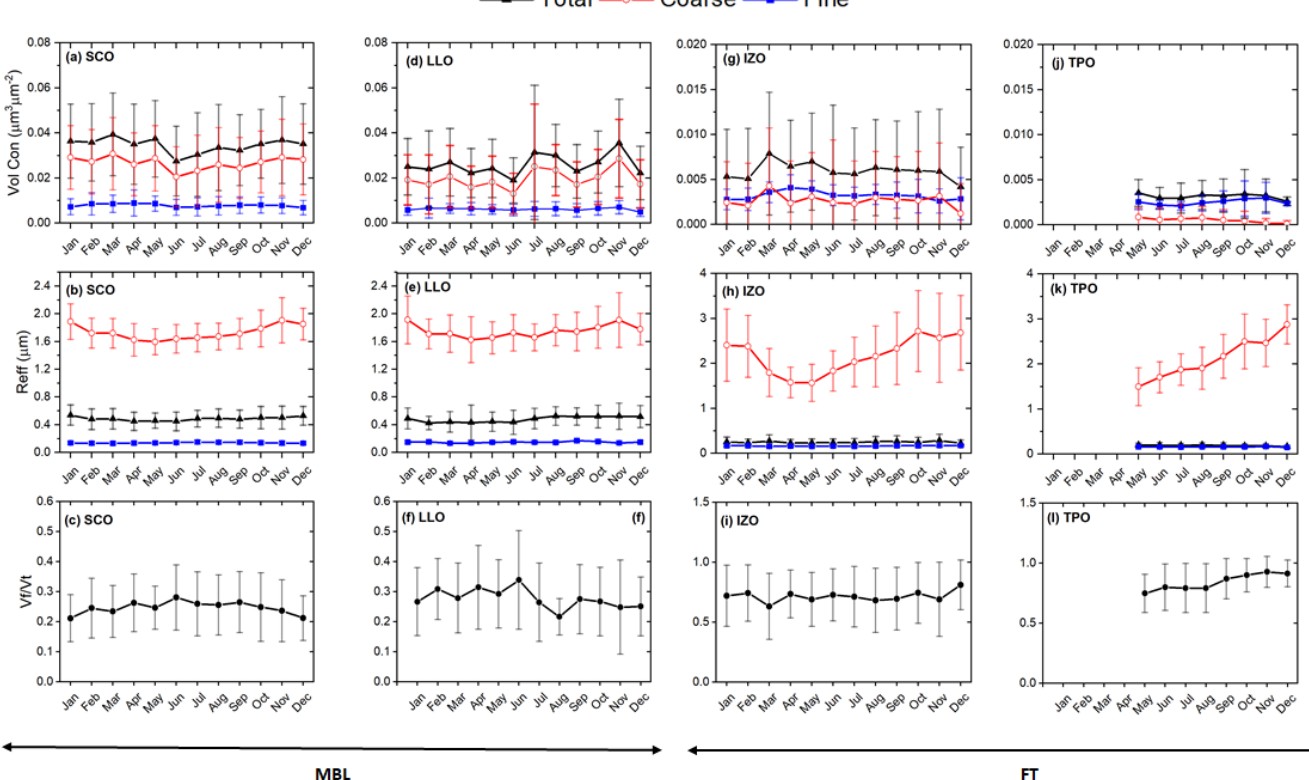

**Figure 6.** Monthly mean volume particle concentration ($VolCon$; $\mu m^3 \ \mu m^{-2}$) and effective radius ($Reff$; $\mu m$) of the total (black color), coarse (red color) and fine (blue color) modes and fine mode volume fraction ($Vf/Vt$) at SCO ((a), (b) and (c)), at LLO ((d), (e) and (f)), at IZO ((g), (h) and (i)) and at TPO ((j), (k) and (l)) under background conditions. The error bars indicate the standard deviation. Note the different scales used in the stations within the MABL and the FT.

bimodal particle size distribution in the MABL with a dominant coarse mode is clearly observed throughout the year in Fig. 7
and Fig. 8 (a) and (b), for SCO and LLO, respectively. This coarse mode is centred at $1.58\pm0.12 \ \mu m$ according to Figs. 8 (b) and (e), with quite consistent values observed at the two stations within the MABL. High aerosol loading was measured in summer (July and August), in some transition months (March and April) and in winter (January and February). These relatively high values correspond to the presence of desert dust in the MABL as a result of the dust transport over this region at higher levels in summer and at lower levels (below 2 km) in winter (Rodríguez et al., 2011; Barreto et al., 2022). It is important to realize
that dust transport in summer provides slightly higher aerosol content (maximum coarse $VolCon$ values of $0.26\pm0.23$ and $0.27\pm0.24 \ \mu m^3 \cdot \mu m^{-2}$ measured in June in SCO and LLO, respectively) than dust transport in autumn-wintertime (minimum coarse $VolCon$ values of $0.12\pm0.08$ and $0.09\pm0.06$ measured in October and January, in SCO and LLO, respectively). These results are consistent with the AOD and AE analysis previously described in Sect. 3.1.

A lower contribution of fine mode aerosols ($Vf/Vt$ of 0.12±0.03) is observed in this case, corroborating more pronounced coarse mode in this dust scenario in comparison to MABL background marine conditions (Dubovik et al., 2002). Nearly constant $Reff$s of 0.12±0.02 $\mu m$ and 1.58±0.12 $\mu m$ for fine and coarse fraction, respectively, have been found to be representative of MABL dust-laden conditions throughout the year. Similarly to $Vf/Vt$, fine and coarse $Reff$s are lower than the values found for clean marine aerosols in Sect. 3.2.1 and also lower than the values of 0.17 $\mu m$ and 1.73 $\mu m$ found for fine and coarse mode, respectively, for pure Saharan dust in the SAMUM-2 field campaign (Toledano et al., 2011). This observed decrease in the effective radius of aerosols in dust-laden conditions is attributed to an effective mixture of mineral dust and marine aerosols.

Size distributions in the FT display a similar seasonal pattern than in the MABL, a consequence of the dust transport pattern over this region. Similar bimodal size distributions are observed in Figs. 7 (c) and (d) at the two sites, with a more prominent coarse mode in summer and spring months centred at 1.57±0.14 $\mu m$, similar to the results presented in Smirnov et al. (1998). These results can be explained by the presence of the SAL as an elevated layer (up to 6 km) in summer, and the preferred low altitude transport of dust in winter (below 2 km) which can also impact sporadically the FT (Barreto et al., 2022). This pattern is corroborated by means of $VolCon$ values with maximum values of 0.16 ±0.12$\mu m^3 \cdot \mu m^{-2}$ in summer in IZO, and 0.06±0.05 $\mu m^3 \cdot \mu m^{-2}$ for the same season in TPO. Similar $Vf/Vt$ values of 0.12±0.03 to those observed in the MABL were measured in the FT, as a consequence of the effective mixing with altitude within the SAL (Barreto et al., 2022). However, these $Vf/Vt$ values are considerably lower than those measured in the FT under background conditions. In this high aerosol loading scenario, $Reff$ is almost constant during the year, with values ranging from 1.50±0.12 $\mu m$ and 1.47±0.13 $\mu m$ for coarse mode aerosols in IZO and TPO, respectively (0.13±0.02 $\mu m$ and 0.14±0.02 $\mu m$ for fine-mode aerosols), which can be attributed to pure desert dust conditions. The coarse mode aerosols in the FT have a lower effective radius than the values of 1.73 $\mu m$ measured by Toledano et al. (2011) in the SAMUM-2 campaign in Cape Verde (January and February 2008) although these data correspond to a specific campaign representative of the tropical dust transport in winter, when the SAL intrudes into the MABL. In this case, gravitational settlement plays a minor role in comparison to its impact during the Saharan intrusions that take place at much higher altitudes in summer over the Canary Islands. Other authors, such as Nakajima et al. (2020, and references therein), suggest the possible underestimation of the coarse aerosols, especially for severe dust storms, as a consequence of the a priori constraint (very low dV(r)/dlnr) introduced in the AERONET inversion procedure for aerosols larger than 10 $\mu m$. However, our findings are quite similar to the particle size distribution measured with in-situ techniques at IZO by Rodríguez et al. (2011) during dust episodes. These authors found two main modes, a fine mode fraction (radius of ~0.1 $\mu m$) attributed to ammonium sulphate, and a coarse mode (radius of ~1.5 $\mu m$), attributed to Saharan advected dust.

No marked seasonal dependence can be observed from the different inversion products presented in Fig. 9. Annually averaged values are presented in Table 1. This lack of seasonal change in this scenario is hypothesised to be due to the stable conditions within the SAL in terms of thermodynamic and aerosol composition already reported by other authors (Prospero and Carlson, 1980; Carlson, 2016; Barreto et al., 2022).

The MABL appears, at both SCO and LLO, as a layer with an average asymmetry parameter maximum at 440 nm ($g$ of 0.77±0.03) and quite constant in the rest of wavelengths ($g$ of 0.75-0.76). These values are in agreement with the results

reported by Dubovik et al. (2002) for a mixture of desert dust and marine aerosols in Cape Verde, in the tropical Eastern North Atlantic. Regarding SSA, aerosols in the MABL are characterised by increasing SSA with wavelength, with maximum values at 440 nm of 0.93±0.03 and 0.95±0.04 in SCO and LLO, respectively. SSA values in the visible and near-infrared seem to be neutral with $\lambda$, with average values of 0.98±0.01 at the two stations. This pattern is coherent with the results expected for a scenario dominated by dust (large) particles (Dubovik et al., 2002). The real part of the refractive index ($n$) adds important additional information on the aerosol's scattering properties while information from the aerosol chemical composition can be inferred from the imaginary part of the refractive index ($k$). Average $n(\lambda)$ of 1.45±0.03 is retrieved for SCO and LLO, with average $k(\lambda)$ values of 0.004±0.002 (maximum values at 440 nm). These results are similar to the values reported by Dubovik et al. (2006, and references therein) and consistent between the two stations.

The two stations in the dust-laden FT, despite the low data availability, especially at TPO, exhibit a similar pattern in terms of the four variables. In these stations, the spectral dependence of $g$ seems to be reduced ($\sim$0.74± 0.01 for the four wavelengths). Regarding the SSA, rather similar values (0.94-0.95) to those found in LLO were reported. These results agree well with the in-situ $SSA$ (0.95) and $g$ (0.74–0.81) which were measured at IZO by polar aerosol photometry on filter samples using a light source resembling the solar spectrum (Kandler et al., 2007). Average $n(\lambda)$ and $k(\lambda)$ of 1.46±0.01 and 0.002±0.001 were found, consistent with the values reported in the literature for desert dust conditions. An average complex refractive index of 1.59–7·10$^{-3}$i was obtained at IZO from the mineralogical model composition derived by electron microscopy (Kandler et al., 2007), which is in excellent agreement with direct optical measurements.

An important point to highlight from these results is the lower impact of dust transport on TPO. Following Barreto et al. (2022), the maximum aerosol loading within the summer SAL is located at an altitude of $\sim$2.5 km slightly higher than the IZO level (2.4 km), with the aerosol extinction decreasing from this altitude. The winter SAL transports dust at lower altitudes than in summer (up to about 2 km height), in the limit of IZO level, and therefore in this season the dust-laden SAL is not expected to reach the TPO altitude. We observe a decrease in total $VolCon$ between IZO and TPO of 0.09 $\mu m^3 \cdot \mu m^{-2}$ (Fig. 8), which corresponds to an average reduction of 43.7%. This reduction is also consistent in terms of average AOD in dust-laden conditions (59.2%) and also considering a regression analysis of AOD difference between IZO and TPO against AOD at IZO, as the reference. In this analysis (Fig. S1 in the supplementary material), a slope of 0.323 and a Pearson coefficient ($R$) of 0.75 are found. With these three pieces of information, we can estimate that the aerosol loading at IZO level is double that of the measured value in the 1-km layer above, at TPO.

### 3.3 Long-Term trends

For the determination of possible trends in the AOD series in both the MABL and the FT, the data of the main stations in each of these layers, SCO and IZO, will be used, as they are the stations with the longest and most complete data series. The time series of monthly mean total-, fine- and coarse-mode AOD at these two stations during a period of 15 and 16 years, respectively, have been deseasonalized by subtracting the mean monthly value of the corresponding month considering all the available years.

The total-, fine- and coarse-mode AOD at IZO, and the total- and coarse-mode AOD at SCO show no trend in the whole period (shown in the supplementary material), in agreement with results obtained by Li et al. (2014). However, the fine-mode

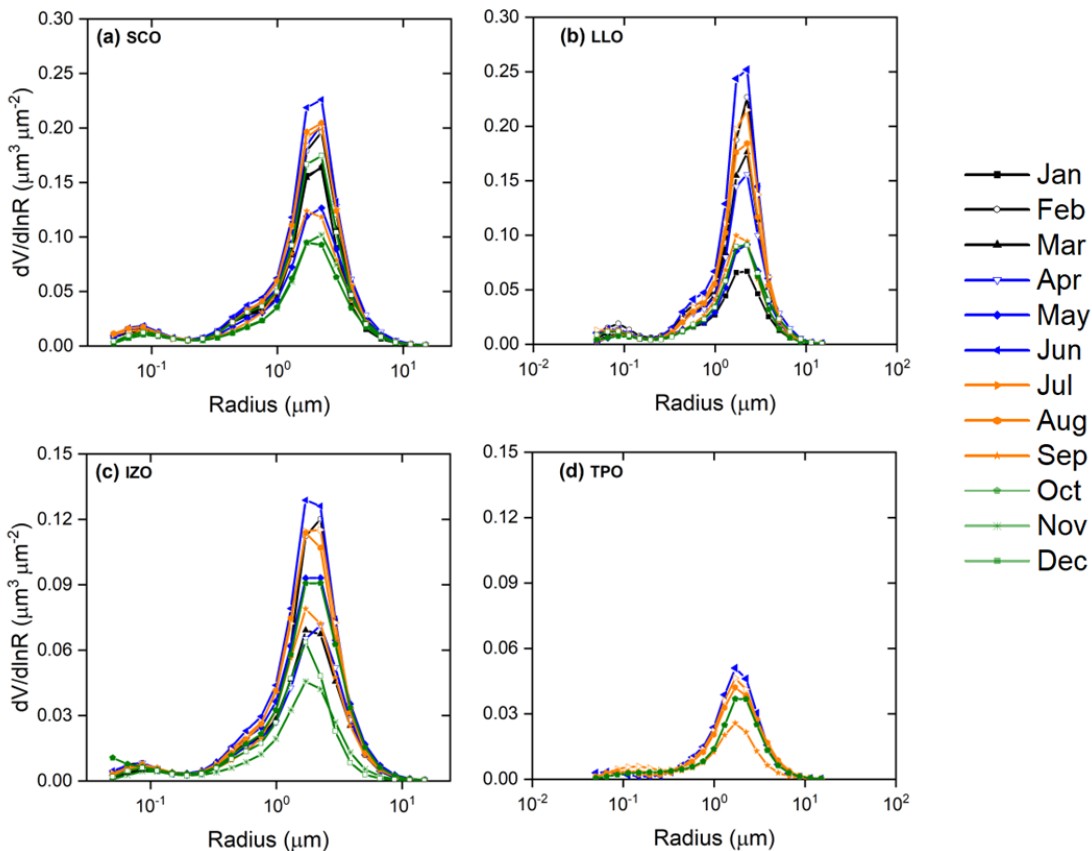

**Figure 7.** Monthly mean aerosol particle size distribution at a) SCO, b) LLO, c) IZO and d) TPO under dust-laden conditions.

AOD at SCO decreases during the study period (Fig. 10), with a trend of $-1.8\pm0.5\cdot10^{-5}$ yr$^{-1}$ (pval $<<$ 0.01). By using Lanzante's method (Lanzante, 1996) on the monthly mean fine-mode AOD values, we confirm one change-point in August 2012. Although this discontinuity is significant at 95% of confidence level, in these periods the time series show no significant drifts. We attribute this change-point to the cease of crude oil refining operations of Santa Cruz de Tenerife refinery (Milford et al., 2018).

As stated in Section 2.1, SCO station is located in Santa Cruz de Tenerife, the capital of Tenerife, a city affected by a complex mixture of anthropogenic sources of pollutants (both on-road and maritime traffic and industrial emissions from an oil refinery) (Milford et al., 2020). The crude oil refinery is located at the SW of the city, at about 3 km far from SCO. The impact of the refinery emissions is maximized in the 10:00–17:00 GMT period due to the effects of meteorology and photochemistry (González and Rodríguez, 2013) coinciding with maximum heating and vertical mixing within the MABL. González and Rodríguez (2013) found that ultrafine particulate concentrations were more sensitive to the fresh emissions of the sources than PM2.5, which was mostly linked to aged fine particles (0.1–1 $\mu m$) of the urban background.

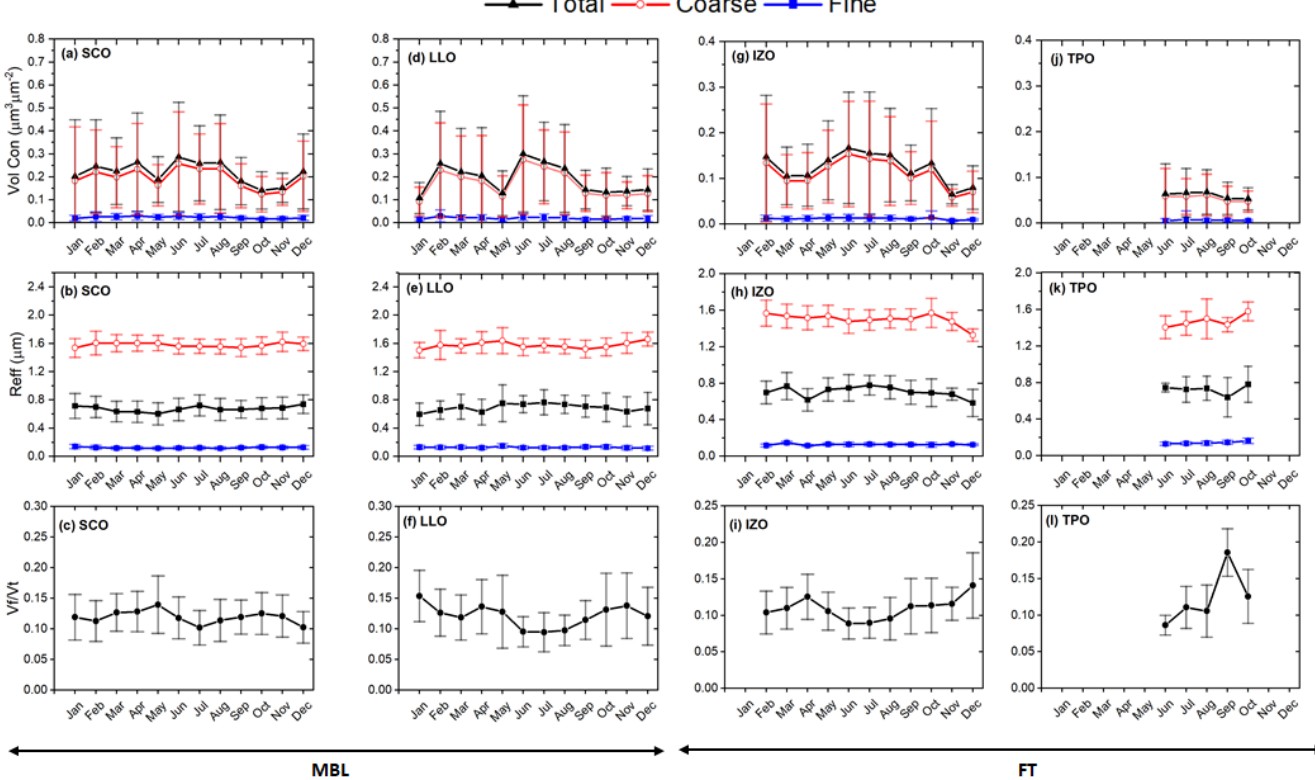

**Figure 8.** Monthly mean volume particle concentration ($VolCon$; $\mu m^3 \ \mu m^{-2}$) and effective radius ($Reff$; $\mu m$) of the total (black color), coarse (red color) and fine (blue color) modes and fine mode volume fraction ($Vf/Vt$) at (a), (b), (c), (d) SCO, (e), (f), (g), (h) LLO, (i), (j), (k), (l) IZO and (m), (n), (o), (p) TPO under dust-laden conditions. The error bars indicate the standard deviation.

Given that during the study period there are no continuous records of fine-mode particulate matter that allow us to detect its changes over time as a consequence of changes in refinery emissions, and compare them with those of the fine-mode AOD series, we have used the $SO_2$ concentrations series measured in the city as a proxy of the temporal evolution of fine-mode particulate matter resulting from refinery emissions. Hourly ambient concentrations of $SO_2$ at Tome Cano station, located in the centre of the city at around 1.5 km from SCO, were obtained from the ambient Air Quality Monitoring Network of

the Canary Islands Government ( https://www3.gobiernodecanarias.org/medioambiente/calidaddelaire/inicio.do, last access: 22 March 2022).

     The 2005-2020 monthly mean $SO_2$ series shows two break points (Fig. 10) according to Lanzante (1996) method. The first of the break points occurs in February 2009, there is an increase in the average annual concentrations of $SO_2$ during the period 2005-2008 and a subsequent decrease following this date. The second break point took place in February 2013, coinciding

with a sharp decline in the oil refining activity. Although the fine-mode AOD series shows a change in behavior after 2008, no break point is detected at this date. We can confirm that between the period 2005-2008, 2009-2013 and 2013-2020, mean

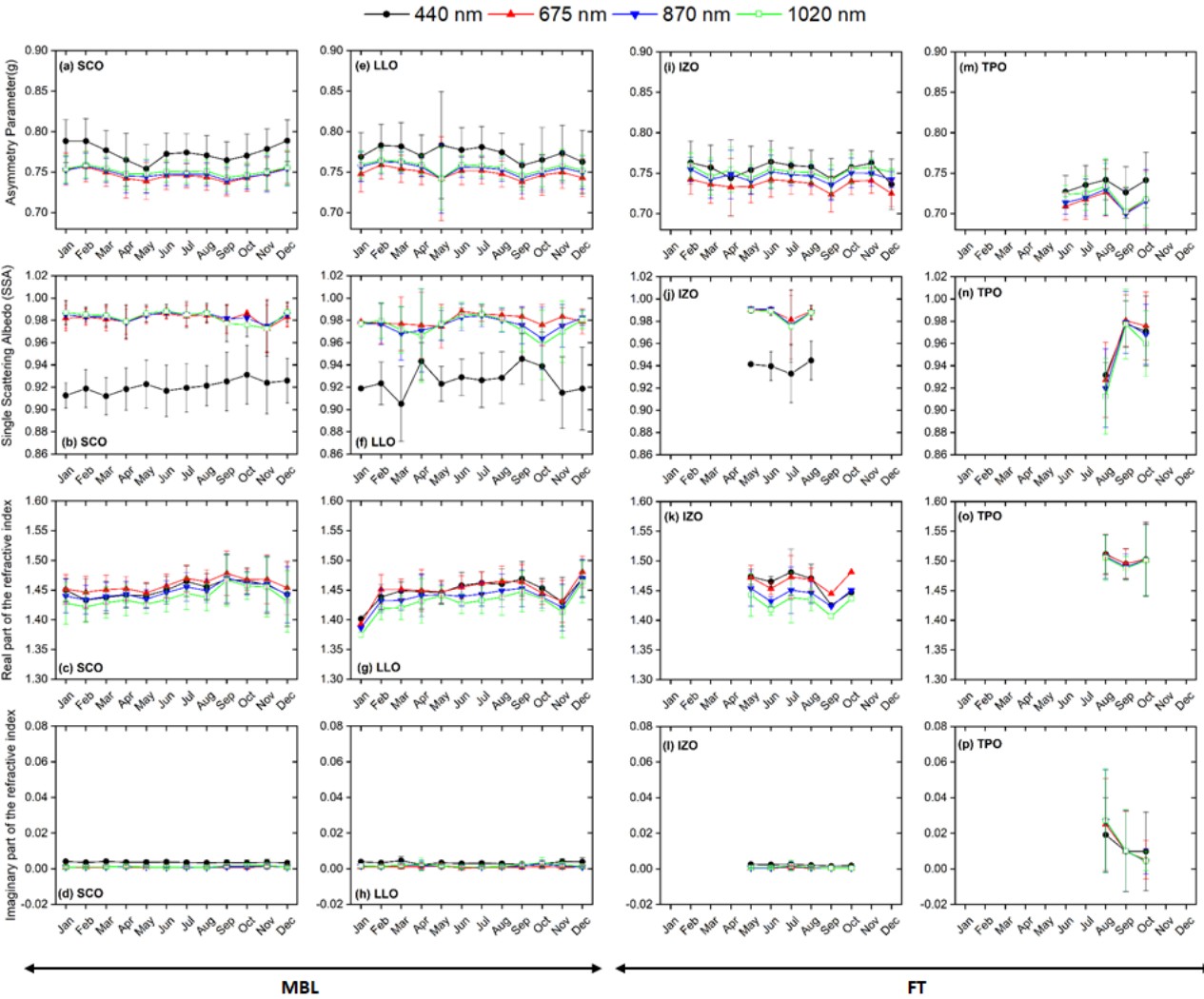

**Figure 9.** Monthly mean of asymmetry parameter ($g$), single scattering albedo ($SSA$), real ($n$) and imaginary ($k$) parts of the refractive index at 440 (black color), 675 (red color), 870 (blue color) and 1020 nm (green color) at SCO (a)-(d), at LLO (e)-(h), at IZO (i)-(l) and at TPO (m)-(p), under dust-laden conditions. Error bars indicates the standard deviation.

**Table 1.** Annual mean of aerosol properties at 440, 675, 870 and 1020 nm of $g$: asymmetry parameter, $SSA$: albedo scattering simple and $n$ and $k$: values of real and imaginary parts of the refractive index at SCO, LLO, IZO and TPO.

| | Wavelength(nm) | SCO | LLO | IZO | TPO |
|---|---|---|---|---|---|
| $<g>$ | 440 | 0.77±0.03 | 0.77±0.03 | 0.76±0.02 | 0.74±0.03 |
| | 675 | 0.75±0.02 | 0.75±0.02 | 0.74±0.02 | 0.72±0.03 |
| | 870 | 0.75±0.02 | 0.75±0.02 | 0.75±0.02 | 0.72±0.03 |
| | 1020 | 0.75±0.02 | 0.76±0.02 | 0.75±0.02 | 0.72±0.03 |
| $<SSA>$ | 440 | 0.92±0.02 | 0.93±0.03 | 0.94±0.02 | 0.95±0.09 |
| | 675 | 0.98±0.01 | 0.98±0.01 | 0.99±0.01 | 0.95±0.10 |
| | 870 | 0.98±0.01 | 0.98±0.02 | 0.98±0.02 | 0.94±0.10 |
| | 1020 | 0.98±0.01 | 0.98±0.02 | 0.98±0.02 | 0.94±0.10 |
| $<n>$ | 440 | 1.455±0.030 | 1.458±0.024 | 1.473±0.030 | 1.505±0.039 |
| | 675 | 1.463±0.026 | 1.458±0.024 | 1.469±0.026 | 1.503±0.040 |
| | 870 | 1.450±0.028 | 1.443±0.025 | 1.446±0.028 | 1.500±0.041 |
| | 1020 | 1.440±0.031 | 1.433±0.027 | 1.434±0.030 | 1.498±0.043 |
| $<k>$ | 440 | 0.004±0.001 | 0.005±0.017 | 0.002±0.001 | 0.015±0.033 |
| | 675 | 0.001±0.001 | 0.002±0.007 | 0.001±0.001 | 0.017±0.043 |
| | 870 | 0.001±0.001 | 0.003±0.007 | 0.001±0.002 | 0.048±0.045 |
| | 1020 | 0.001±0.001 | 0.003±0.007 | 0.001±0.002 | 0.018±0.045 |

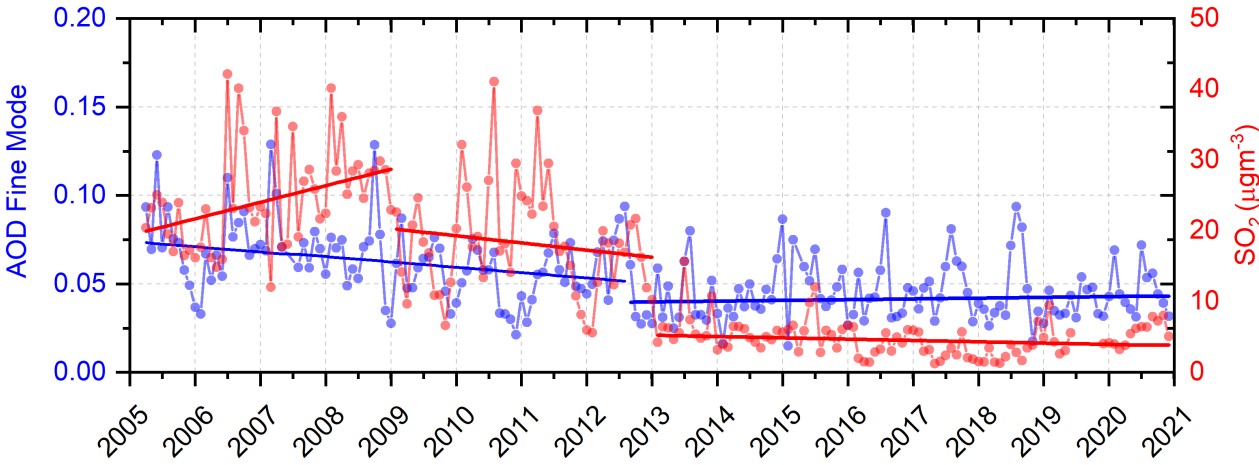

**Figure 10.** Series of monthly mean values of fine-mode AOD (SCO, in blue) and $SO_2$ (Tomé Cano station, in red) in the study period 2005-2020.

$SO_2$ concentrations of 24.3±0.02, 18.2±1.2 and 4.5±0.4 $\mu gm^{-3}$, respectively, were recorded, corresponding to a decrease of 81% in the 2013-2020 period compared to the 2005-2008 period. This decrease observed in $SO_2$ concentrations since 2013 is in response to the large reduction of oil refining $SO_2$ emissions in the city (Milford et al., 2018). This decrease in the concentration of $SO_2$ corresponds to a decrease in fine-mode AOD of 57%.

## 4 Summary and Conclusions

Long-term ground-based AERONET observations have been used in this study to perform a comprehensive characterization of atmospheric aerosols in the Subtropical Eastern North Atlantic. Photometric measurements from four stations with an extensive temporal (9 to 16 years) and vertical coverage (from the sea level to 3555 m height) allow us to perform a robust assessment of
tropospheric aerosols in this subtropical region. Santa Cruz de Tenerife –SCO- and La Laguna –LLO- are stations within the MABL while the other two stations, Izaña –IZO- and Teide Peak –TPO-, are high mountain stations within the FT.

The MABL and FT AOD and AE aerosol characterization performed in this study confirm the predominance of the alternating situations in this region, from background to dust-loaded Saharan air mass outbreaks seasonally affecting the four sites as a result of the seasonal dust transport over the Subtropical North Atlantic. Background conditions prevail in the MABL most
of the year, particularly in May and June ($> 15$ days per month) while dust-laden conditions dominate in July and August. Regarding the FT, dust-laden conditions follow a seasonal pattern with background FT conditions dominating most of the year ($> 50\%$ of days every month) except in July and August, with a similar number of days under dust-laden and background conditions.

Under MABL background conditions, a bimodal lognormal size distribution with a predominant coarse mode, relatively
stable and low $VolCon$ values (between $0.02\pm0.01$ $\mu m^3 \cdot \mu m^{-2}$ and $0.04\pm0.02$ $\mu m^3 \cdot \mu m^{-2}$) and a quite consistent analysis in terms of $Reff$ have been found in the two stations within the MABL throughout the year (aerosols in the coarse mode with $Reff$ ranging from $1.60\pm 0.19$ $\mu m$ in late spring-early summer to $1.91\pm0.34$ $\mu m$ in winter). These results confirm the effective vertical mixing within the cloud-free subtropical MABL (Carrillo et al., 2016; Barreto et al., 2022). Regarding the FT, background conditions with remarkably low aerosol loading characterised by a slight bimodality of the aerosol particle size
distribution are observed at the two high-mountain stations, with a predominant impact of fine-mode aerosols throughout the year ($Vf/Vt$ with a maximum value of $0.93\pm0.13$ in November), with an average $Reff$ of $0.16\pm0.02$ $\mu m$.

Dust-laden conditions are characterized by a bimodal particle size distribution in the MABL with a dominant coarse mode observed throughout the year centred at $1.58 \pm0.12 \mu m$. Maximum aerosol loading was measured in summer (July and August) (maximum $VolCon$ of $0.27\pm0.24$ $\mu m^3 \cdot \mu m^{-2}$) as a result of the dust transport over the Saharan convective boundary layer
at higher levels in summer, a structure more prevalent in the free troposphere (Reid et al., 2003), which can often extend to 5-6 km height (Prospero and Carlson, 1972, 1980; Carlson, 2016; Barreto et al., 2022). Our results indicate that aerosol concentration in the MABL under dust-laden conditions is almost one order of magnitude higher in comparison to marine background conditions, with a lower contribution of fine mode aerosols ($Vf/Vt$ of $0.12\pm0.03$) and nearly constant $Reff$s of $0.12\pm0.02$ $\mu m$ and $1.58\pm0.12$ $\mu m$ for fine and coarse fraction. Furthermore, coarse $Reff$s are lower than the value of
$1.73\pm0.23$ and $1.75 \pm 0.27$ $\mu m$ found for clean marine aerosols. Quite consistent values are also observed at the two stations within the MABL. A seasonal pattern in the aerosol volume size distribution is also observed in the FT as a consequence of the dust transport pattern over this region. Bimodal size distributions with a more prominent coarse mode in summer and spring months centred at $1.57\pm0.14$ $\mu m$ are observed at the two sites, with maximum $VolCon$ values of $0.16\pm0.12$ $\mu m^3 \cdot \mu m^{-2}$ in summer. Similar $Vf/Vt$ values of $0.12\pm0.03$ to those observed in the MABL were measured in the FT, but considerably

lower than those measured in the FT under background conditions. $Reff$ is observed to be almost constant during the year, with values ranging from 1.50±0.12 $\mu m$ and 1.47±0.13 $\mu m$, which can be attributed to pure desert dust conditions in the Subtropical North Atlantic.

The lack of seasonal dependence observed in the rest of inversion products ($g$, $SSA$, $n$ and $k$) in the dust-laden scenario is hypothesised to be due to the stable conditions within the SAL in terms of thermodynamic and aerosol composition. The MABL appears as a layer with an average asymmetry parameter minimum at 440 nm ($g$ of 0.77±0.03), with wavelength independent SSA values in the visible and near-infrared, with average values of 0.98±0.01 at the two stations (0.93-0.95 for 440 nm). Average $n(\lambda)$ values of $1.45 \pm 0.03$ and $k(\lambda)$ values of 0.004±0.002 (maximum values at 440 nm) are retrieved to be representative of the MABL. Spectral dependence of $g$ seems to be reduced in the FT, with values of $0.74 \pm 0.01$ for the four wavelengths. Regarding the SSA, rather similar values to those found in the MABL were reported. Average $n$ and $k$ of 1.46±0.01 and 0.002±0.001 were found, respectively.

The low impact of dust transport at TPO level is an important outcome of this paper, with an observed decrease in the aerosol concentration at TPO ranging from 32.3% to 59.2%. With this information, we have estimated the aerosol loading at IZO level to be double the aerosol concentration in the 1-km layer above, at TPO.

A subsequent long-term trend analysis in Santa Cruz over a 15-year period showed a negative trend in fine-mode AOD, with a trend of -1.8±0.5·$10^{-5}$ yr$^{-1}$ (pval $<< 0.01$). No significant trend was observed in the total- and coarse-mode AOD in SCO as well as in the total-, fine- and coarse-mode AOD at IZO in the whole period. The $SO_2$ concentration time series at SCO was used as a proxy of the temporal evolution of fine-mode particulate matter resulting from refinery emissions. This information helped us to link the decrease in the concentration of $SO_2$ in response to the large reduction of oil refining $SO_2$ emissions at Santa Cruz refinery to the decrease observed in the fine-mode AOD, estimated at 57%. These results suggest that AERONET AOD observations, and specifically the fine-mode AOD, appear to be sensitive enough to detect long-term changes in air quality in a city.

The aerosol characterization performed in this paper has the potential to provide a wide set of aerosol properties relevant for climate studies in a region that can be considered a key location to study the seasonal dependence in the dust transport from the Sahel-Sahara to the Caribbean region. This is a robust characterization of the MABL and FT by means of a consistent analysis of the four stations under different and contrasting aerosols regimes, including background marine, pure Saharan dust and the very stable and low aerosol turbidity within the FT. Such observations can be useful to study long-term trends in atmospheric composition within the MABL, changes affecting the FT, considered representative of large areas avoiding possible contamination from local or regional sources, for validating aerosol models or to properly constrain pre-defined parameters in current inversion schemes.

**Table A1.** Monthly mean, median and std of aerosol optical depth (AOD) at 440, 500, 675 and 870 nm and Ångström exponent ($AE_{440-870nm}$) at SCO from April 2005 and December 2020 and LLO between July 2006 and December 2020. N is number of data.

| | Month | N | AOD(440 nm) | | | AOD(500 nm) | | | AOD(675 nm) | | | AOD(870 nm) | | | AE | | |
|---|---|---|---|---|---|---|---|---|---|---|---|---|---|---|---|---|---|
| | | | Mean | Median | Std | Mean | Median | Std | Mean | Median | Std | Mean | Median | Std | Mean | Median | Std |
| | Jan | 313 | 0.13 | 0.09 | 0.16 | 0.12 | 0.08 | 0.16 | 0.10 | 0.07 | 0.16 | 0.09 | 0.06 | 0.15 | 0.62 | 0.57 | 0.35 |
| | Feb | 277 | 0.14 | 0.09 | 0.16 | 0.13 | 0.09 | 0.16 | 0.12 | 0.07 | 0.15 | 0.11 | 0.07 | 0.14 | 0.55 | 0.46 | 0.36 |
| | Mar | 308 | 0.17 | 0.11 | 0.16 | 0.16 | 0.10 | 0.16 | 0.14 | 0.09 | 0.16 | 0.13 | 0.08 | 0.15 | 0.54 | 0.48 | 0.30 |
| | Apr | 318 | 0.15 | 0.11 | 0.17 | 0.14 | 0.10 | 0.17 | 0.12 | 0.07 | 0.16 | 0.11 | 0.07 | 0.15 | 0.67 | 0.66 | 0.30 |
| | May | 380 | 0.13 | 0.10 | 0.10 | 0.12 | 0.09 | 0.09 | 0.10 | 0.07 | 0.09 | 0.09 | 0.07 | 0.08 | 0.70 | 0.69 | 0.26 |
| **SCO** | Jun | 382 | 0.16 | 0.09 | 0.20 | 0.15 | 0.08 | 0.20 | 0.13 | 0.06 | 0.20 | 0.12 | 0.05 | 0.19 | 0.70 | 0.68 | 0.35 |
| | Jul | 395 | 0.25 | 0.16 | 0.23 | 0.25 | 0.16 | 0.23 | 0.23 | 0.14 | 0.22 | 0.22 | 0.13 | 0.21 | 0.45 | 0.32 | 0.34 |
| | Aug | 417 | 0.27 | 0.20 | 0.25 | 0.26 | 0.20 | 0.25 | 0.24 | 0.17 | 0.25 | 0.23 | 0.16 | 0.24 | 0.44 | 0.34 | 0.31 |
| | Sep | 414 | 0.16 | 0.12 | 0.12 | 0.15 | 0.11 | 0.12 | 0.13 | 0.09 | 0.12 | 0.12 | 0.08 | 0.11 | 0.65 | 0.61 | 0.34 |
| | Oct | 365 | 0.14 | 0.11 | 0.12 | 0.13 | 0.10 | 0.12 | 0.10 | 0.08 | 0.11 | 0.09 | 0.07 | 0.10 | 0.71 | 0.67 | 0.36 |
| | Nov | 289 | 0.13 | 0.10 | 0.09 | 0.12 | 0.09 | 0.08 | 0.10 | 0.08 | 0.08 | 0.09 | 0.07 | 0.07 | 0.59 | 0.51 | 0.33 |
| | Dic | 303 | 0.13 | 0.09 | 0.13 | 0.12 | 0.09 | 0.13 | 0.10 | 0.07 | 0.12 | 0.10 | 0.06 | 0.12 | 0.61 | 0.54 | 0.34 |
| | Total | – | 0.16 | 0.12 | 0.16 | 0.15 | 0.11 | 0.16 | 0.13 | 0.09 | 0.15 | 0.12 | 0.08 | 0.14 | 0.60 | 0.55 | 0.33 |
| | Jan | 164 | 0.10 | 0.08 | 0.07 | 0.09 | 0.07 | 0.07 | 0.07 | 0.05 | 0.07 | 0.06 | 0.04 | 0.06 | 0.76 | 0.76 | 0.35 |
| | Feb | 145 | 0.13 | 0.08 | 0.16 | 0.12 | 0.08 | 0.16 | 0.11 | 0.06 | 0.15 | 0.10 | 0.05 | 0.14 | 0.65 | 0.58 | 0.35 |
| | Mar | 172 | 0.16 | 0.10 | 0.17 | 0.15 | 0.09 | 0.17 | 0.14 | 0.07 | 0.16 | 0.12 | 0.06 | 0.15 | 0.62 | 0.57 | 0.35 |
| | Apr | 183 | 0.13 | 0.09 | 0.15 | 0.12 | 0.08 | 0.15 | 0.10 | 0.06 | 0.14 | 0.09 | 0.05 | 0.14 | 0.84 | 0.85 | 0.38 |
| | May | 198 | 0.10 | 0.08 | 0.07 | 0.09 | 0.07 | 0.07 | 0.07 | 0.05 | 0.07 | 0.06 | 0.04 | 0.06 | 0.89 | 0.93 | 0.32 |
| **LLO** | Jun | 192 | 0.17 | 0.08 | 0.24 | 0.16 | 0.07 | 0.25 | 0.14 | 0.05 | 0.24 | 0.13 | 0.04 | 0.24 | 0.81 | 0.81 | 0.49 |
| | Jul | 309 | 0.27 | 0.21 | 0.23 | 0.26 | 0.20 | 0.23 | 0.25 | 0.19 | 0.22 | 0.23 | 0.18 | 0.21 | 0.42 | 0.25 | 0.37 |
| | Aug | 271 | 0.25 | 0.19 | 0.25 | 0.25 | 0.18 | 0.25 | 0.23 | 0.16 | 0.25 | 0.22 | 0.15 | 0.24 | 0.41 | 0.27 | 0.34 |
| | Sep | 237 | 0.14 | 0.11 | 0.12 | 0.13 | 0.09 | 0.12 | 0.12 | 0.07 | 0.12 | 0.10 | 0.06 | 0.11 | 0.68 | 0.66 | 0.37 |
| | Oct | 181 | 0.11 | 0.09 | 0.09 | 0.10 | 0.08 | 0.09 | 0.08 | 0.06 | 0.08 | 0.07 | 0.05 | 0.08 | 0.74 | 0.71 | 0.34 |
| | Nov | 169 | 0.11 | 0.09 | 0.08 | 0.10 | 0.08 | 0.07 | 0.09 | 0.06 | 0.07 | 0.08 | 0.06 | 0.06 | 0.68 | 0.64 | 0.31 |
| | Dec | 160 | 0.11 | 0.08 | 0.10 | 0.10 | 0.07 | 0.10 | 0.08 | 0.06 | 0.10 | 0.07 | 0.05 | 0.09 | 0.74 | 0.72 | 0.38 |
| | Total | – | 0.15 | 0.11 | 0.14 | 0.14 | 0.10 | 0.14 | 0.12 | 0.08 | 0.14 | 0.11 | 0.07 | 0.13 | 0.69 | 0.65 | 0.36 |

**Table B1.** Monthly mean, median and std of aerosol optical depth (AOD) at 440, 500, 675 and 870 nm and Ångström exponent ($AE_{440-870nm}$) at IZO between from October 2004 and December 2020 and TPO between July 2012 and December 2020. N is number of data.

| | Month | N | AOD(440 nm) Mean | Median | Std | AOD(500 nm) Mean | Median | Std | AOD(675 nm) Mean | Median | Std | AOD(870 nm) Mean | Median | Std | AE Mean | Median | Std |
|---|---|---|---|---|---|---|---|---|---|---|---|---|---|---|---|---|---|
| **IZO** | Jan | 341 | 0.03 | 0.02 | 0.07 | 0.03 | 0.01 | 0.07 | 0.02 | 0.01 | 0.06 | 0.02 | 0.01 | 0.06 | 1.03 | 1.06 | 0.30 |
| | Feb | 287 | 0.03 | 0.02 | 0.05 | 0.03 | 0.02 | 0.05 | 0.02 | 0.01 | 0.04 | 0.02 | 0.01 | 0.04 | 0.98 | 1.01 | 0.34 |
| | Mar | 356 | 0.04 | 0.03 | 0.06 | 0.04 | 0.02 | 0.06 | 0.03 | 0.01 | 0.06 | 0.03 | 0.01 | 0.05 | 0.96 | 1.03 | 0.38 |
| | Apr | 404 | 0.05 | 0.03 | 0.06 | 0.04 | 0.03 | 0.05 | 0.03 | 0.02 | 0.05 | 0.03 | 0.01 | 0.05 | 1.07 | 1.14 | 0.34 |
| | May | 458 | 0.04 | 0.03 | 0.05 | 0.04 | 0.03 | 0.05 | 0.03 | 0.02 | 0.05 | 0.03 | 0.01 | 0.05 | 1.03 | 1.10 | 0.32 |
| | Jun | 443 | 0.06 | 0.02 | 0.10 | 0.06 | 0.02 | 0.10 | 0.05 | 0.01 | 0.10 | 0.05 | 0.01 | 0.09 | 0.93 | 1.04 | 0.41 |
| | Jul | 464 | 0.15 | 0.09 | 0.16 | 0.15 | 0.09 | 0.16 | 0.14 | 0.08 | 0.15 | 0.13 | 0.08 | 0.15 | 0.54 | 0.35 | 0.47 |
| | Aug | 448 | 0.13 | 0.09 | 0.14 | 0.13 | 0.08 | 0.14 | 0.12 | 0.07 | 0.13 | 0.11 | 0.07 | 0.13 | 0.53 | 0.32 | 0.43 |
| | Sep | 392 | 0.07 | 0.03 | 0.08 | 0.06 | 0.03 | 0.08 | 0.05 | 0.02 | 0.08 | 0.05 | 0.02 | 0.08 | 0.81 | 0.84 | 0.44 |
| | Oct | 370 | 0.04 | 0.02 | 0.04 | 0.03 | 0.02 | 0.04 | 0.03 | 0.01 | 0.04 | 0.02 | 0.01 | 0.04 | 0.95 | 1.02 | 0.40 |
| | Nov | 353 | 0.03 | 0.02 | 0.03 | 0.03 | 0.02 | 0.03 | 0.02 | 0.01 | 0.03 | 0.02 | 0.01 | 0.03 | 1.00 | 1.04 | 0.35 |
| | Dec | 360 | 0.03 | 0.02 | 0.03 | 0.02 | 0.01 | 0.03 | 0.02 | 0.01 | 0.02 | 0.01 | 0.01 | 0.02 | 1.08 | 1.11 | 0.31 |
| | Total | – | 0.06 | 0.03 | 0.07 | 0.05 | 0.03 | 0.07 | 0.05 | 0.02 | 0.07 | 0.04 | 0.02 | 0.07 | 0.91 | 0.92 | 0.38 |
| **TPO** | Jan | – | – | – | – | – | – | – | – | – | – | – | – | – | – | – | |
| | Feb | – | – | – | – | – | – | – | – | – | – | – | – | – | – | – | |
| | Mar | – | – | – | – | – | – | – | – | – | – | – | – | – | – | – | |
| | Apr | – | – | – | – | – | – | – | – | – | – | – | – | – | – | – | |
| | May | 33 | 0.02 | 0.02 | 0.01 | 0.02 | 0.02 | 0.01 | 0.01 | 0.01 | 0.01 | 0.01 | 0.01 | 0.00 | 1.21 | 1.24 | 0.16 |
| | Jun | 108 | 0.05 | 0.02 | 0.08 | 0.04 | 0.01 | 0.08 | 0.04 | 0.01 | 0.08 | 0.03 | 0.01 | 0.07 | 0.99 | 1.13 | 0.40 |
| | Jul | 190 | 0.10 | 0.05 | 0.15 | 0.10 | 0.04 | 0.14 | 0.09 | 0.04 | 0.12 | 0.08 | 0.04 | 0.11 | 0.60 | 0.49 | 0.45 |
| | Aug | 197 | 0.08 | 0.04 | 0.09 | 0.07 | 0.03 | 0.09 | 0.07 | 0.03 | 0.09 | 0.06 | 0.03 | 0.08 | 0.63 | 0.50 | 0.42 |
| | Sep | 207 | 0.03 | 0.02 | 0.04 | 0.03 | 0.01 | 0.04 | 0.02 | 0.01 | 0.04 | 0.02 | 0.01 | 0.03 | 1.01 | 1.12 | 0.38 |
| | Oct | 193 | 0.02 | 0.02 | 0.02 | 0.02 | 0.01 | 0.02 | 0.01 | 0.01 | 0.02 | 0.01 | 0.01 | 0.02 | 1.07 | 1.11 | 0.32 |
| | Nov | 67 | 0.02 | 0.02 | 0.01 | 0.02 | 0.01 | 0.01 | 0.01 | 0.01 | 0.01 | 0.01 | 0.01 | 0.01 | 1.16 | 1.21 | 0.23 |
| | Dec | 13 | 0.02 | 0.02 | 0.00 | 0.01 | 0.01 | 0.00 | 0.01 | 0.01 | 0.00 | 0.01 | 0.01 | 0.00 | 1.40 | 1.43 | 0.20 |
| | Total | – | 0.04 | 0.02 | 0.05 | 0.04 | 0.02 | 0.05 | 0.03 | 0.02 | 0.04 | 0.03 | 0.01 | 0.04 | 1.01 | 1.03 | 0.32 |

*Data availability.* The data from AERONET used in the present study can be freely obtained from https://aeronet.gsfc.nasa.gov (Holben et al., 1998). The data from SIMAC (Gobierno de Canarias) can be freely accessed at https://www3.gobiernodecanarias.org/medioambiente/calidaddelaire/inicio.do.

*Author contributions.* C.G and E.C designed the structure and methodology of an early version of the paper. A.B and R.D.G redesigned the
410   structure and methodology to obtain the final paper. R.D.G. computed the calculations required. A.B., R.D.G. E.C., C.M., F.A.. S.L. and C.T. discussed the results and participated in the retrievals analysis. E.C, C.M. and R.D.G wrote section 3.3.. F.E. and J.D. ensured the provision of funds and the operation of the LLO. All authors discussed the results and contributed to the final paper.

*Competing interests.* The authors declare that they have not conflict of interest.

*Acknowledgements.* We gratefully acknowledge the data provided by AERONET network. We wish to express our appreciation to the staffs
415   from AEMET and ULL for maintaining the instrumentation, and ensuring the quality of the data. AERONET sun photometers at Izaña have been calibrated through AEROSPAIN Central Facility (https://aerospain.aemet.es/). This study is a contribution to the Barcelona Dust Forecast Centre (https://dust.aemet.es/). The authors also acknowledge the support from ACTRIS, Ministerio de Ciencia e Innovación from Spain through the projects SYNERA (PID2020- 118793GA-I00) and ePOLAAR (RTI2018-097864-B-I00) and by Junta de Castilla y León (grant no. VA227P20).

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
