# Peer review of "Aerosol characterization in the Subtropical Eastern North Atlantic region derived from long-term AERONET measurements"

_Atmospheric Chemistry and Physics, 2022_

## Author Comment (AC1)

*Aerosol characterization in the Subtropical Eastern North Atlantic region derived from long-term AERONET measurements*

*África Barreto et al.*

**Referee Report #RC1**

**The paper describes and summarizes the columnar aerosol optical characteristics of air masses passing over Tenerife island toward Europe in a long period, ranging from 2005 to 2020. The measurements, taken in the 4 sites located at different levels from the surface, allow a good characterization of the atmospheric layers. Although the paper doesn't present novel concepts or ideas, the produced results are very important for climate studies and also to understand how optical characteristics of air masses coming from Africa modifies during their passage toward Europe.**

*Authors:*  *We acknowledge the referee's positive and constructive comments.*

**The analysis is well structured and clearly explained and the substantial conclusions are reached. The week part, on my opinion, is the section 3.3: Long-Term trends. The supplementary material contains not the trends but the time behavior of AOD -total, fine and coarse- at IZO (between 2005 and the end of 2020, please correct the caption) and AOD -total and coarse at SCO (between 2005 and the end of 2020, please correct the caption). The time behavior of AOD -fine is shown only in Figure 10, together with its trend. It would be better showing AOD -fine in the supplementary plot too. From the analysis it seems that only SCO shows a trend in AOD-fine. Would it be possible to analyze for both SCO and IZO what happen to the absorption capabilities (Absorption Aerosol optical Depth, or SSA) and refractive index? Is there any trend in these parameters in both the sites?**

*Authors:*  *Thank you very much for your comments.*

- *The captions have been corrected.*
- *We have added the AOD -fine time series in the supplementary at SCO.*

[Figure]

*Figure S2.- Series of monthly mean values of total (black), fine (blue) and coarse-mode AOD (red) at SCO between 2006 and 2020.*

- *We have not analysed the trends in the SSA and refractive index time series at SCO and IZO because we do not have enough data in Level 2.0 to do this study (SSA and refractive index at SCO: 293 days, SSA and refractive index at IZO: 45 days).*

**I also have some minor corrections:**

**Abstract:**

**line 4: "this site is can …" please correct;**

*Authors: Done*

**line 16: "and fine mode fraction <0.35" correct with " volume contribution of the fine mode fraction Vf/Vt <0.35"**

*Authors: Done*

**Introduction:**

**Line 50-51: state somewhere the site is Tenerife:**

*Authors: We have added the following information in the text:*

> *"In this study, we describe the long-term seasonal evolution of atmospheric aerosols by using AERONET observations at four different sites at different altitudes **in Tenerife**, in the Subtropical Eastern North Atlantic region**."***

**Line 56 define and provide a ref for "dust belt"**

*Authors: The authors have provided in this sentence (Alonso-Pérez et al., 2007, 2011, 2012; Rodríguez et al., 2011; Cuevas et al., 2015) a suitable number of references for this "dust belt", as the region affected by the mineral dust transport from North Africa. These papers are able to define this region and also the marked seasonal evolution of the dust transport at these latitudes.*

**Section 3.1:**

**Line 152 What about the minimum visible in May?**

*Authors: We agree with this comment. We have changed the values included in the text as follows:*

> *"SCO **(Fig. 2 (a))** and LLO **(Fig. 2 (c))** display low $AOD_{500nm}$ values **in May (0.12±0.09 at SCO and 0.09±0.07 at LLO) and** between October and February, with values of 0.12±0.03 at SCO and 0.10±0.03 at LLO."*

---

## Author Comment (AC2)

**Aerosol characterization in the Subtropical Eastern North Atlantic region derived from long-term AERONET measurements**

África Barreto et al.

Referee Report #RC2

This study from A. Barreto et al. performs a multi-annual characterization of the columnar aerosol properties (AOD, AE, SSA, etc) obtained by AERONET at four sites in the Tenerife island, two of them representative of the marine boundary layer, and the other two representative of the free troposphere.

The location of the island and the instruments is privileged and the length of the different data series is notable (9 to 16 years) so the results are of evident interest for the scientific community. Although the techniques are not novel, they are extensively used, and the data analysis, the presentation and the use of English is good. My recommendation is to publish the article after few corrections listed below. No major general corrections are placed.

Authors: We appreciate the positive and constructive comments of the Referee. Below we respond to his/her specific comments.

Specific comments:

Abstract.

Line 4: "This site is can be..." please correct.

Authors: Done

Line 9: Please write Angstrom the same in all appearances in the text (A^o\_ngström)

Authors: Done

Line 90: I think ground-based is more adequate.

**Authors:** We agree with this comment. We have corrected this mistake in the text.

Lines 119-120: the limited availability of data at TPO is due to bad weather conditions. But, is the instrument removed from the site? I would bet it is, but it could be stated so in the text.

**Authors:** The instrumentation at TPO used to be removed from the station to avoid bad weather conditions. However, since Sept. 2020, the instrument is in operation at the station without interruption. We have included the following information in the text:

"SCO, LLO and IZO are devoted to continuous long-term monitoring. **AERONET** measurements at TPO, due to adverse weather conditions, are mainly available between mid-spring and mid-autumn, **having continuous records from September 2020.**"

**Line 170: What are the AE values recorded in background conditions? (only AOD in background conditions is indicated)**

**Authors:** We agree with this comment. AE values have been included in the manuscript.

"In contrast to these dominating background conditions, higher  $AOD_{500nm}$  and lower  $AE_{440-870nm}$  values are recorded in July ( $AOD_{500nm}$  of  $0.15\pm0.16$  and  $AE_{440-870nm}$  of  $0.54\pm0.47$  at IZO;  $0.10\pm0.14$  and  $0.60\pm0.45$  at TPO) and August ( $0.13\pm0.14$  and  $0.53\pm0.43$  at IZO;  $0.07\pm0.09$  and  $0.63\pm0.42$  at TPO)."

Line 185: the slight increase in March merits a short explanation even if it is extracted from literature.

**Authors:** In the intermediate seasons (spring and fall) the Canaries are in the trajectory of the polar outbreaks or the tropical thunderstorms along the discontinuities between the marine and continental air masses (Bergametti et al., 1989), illustrating the extremely variable climatology in these transition months. This is the explanation of the highly variable conditions found in March-April and September-October.

Bergametti, G., Gomes, L., Coud-Gaussen, G., Rognon, P. and Le Coustumer, M. N.: African dust observed over Canary Islands: Source-regions identification and transport pattern for some summer situations, Journal of Geophysical Research: Atmospheres (94) D12, 14855-14864, 10.1029/JD094iD12p14855, 1989.

Line 206: it is said that the coarse mode in Figure 5 is slightly increased in the summer and spring months in case of LLO, but in the figure it seems the yellow line corresponding to June has the smaller coarse mode, even if summer. Not sure about all the months due to color and symbols. Some of them are hardly visible. Please consider to group them in seasons, or use only selected months, or increase the size of symbols.

**Authors:** The authors agree with this comment. Figs. 5 and 7 have been changed. We have used a code of colours according to the seasonality proposed by the referee, we have changed the symbols and the size of the figures as follows:

Figure 5.- Monthly mean aerosol particle size distribution at a) SCO, b) LLO, c) IZO and d) TPO under background conditions.

---

## Author Comment (AC3)

*Aerosol characterization in the Subtropical Eastern North Atlantic region derived from long-term AERONET measurements*

*África Barreto et al.*

**Referee Report #RC3**

This work of Barreto et al, focus on four very close AERONET stations on Tenerife island. Timeseries are long enough to provide adequate results for the area and the topography makes the place ideal for studying the stratification of Boundary Layer and the effect on the Aerosol optical properties. Additionally, the location is in a region with very frequent Saharan dust intrusions that alters the aerosols regime. The study is of high interest and in it fits the journal. The presentation is very well organized and the results will push forward the scientific knowledge on for the aerosol state above north Atlantic ocean. I suggest to accept it for publication after minor revisions.

*Authors:  The authors would like to thank Referee #3 for the very positive review of this paper and the acceptance for publication.*

**Comments:**

**A lot of commas are missing. Please read carefully and correct the syntax. Eg line 13,40,43.**

*Authors:  The authors agree with this comment. We have added commas in the following sentences:*

*Line 13: "… most of the year, while dust-laden…"*

*Line 41: "… observations, freely available for…"*

*Line 61: "… temperature gradients, strongly affecting…"*

**L66. In the previous sentence the typical stratification of the troposphere in the region is described. This sentence begins with "these contrasting aerosol regimes". I think a sentence is missing linking the layers with aerosol regimes. Also, the characterization of "key site for aerosol monitoring" is repeated 10 lines earlier.**

*Authors:  We agree with this comment. We propose to re-phrase the paragraph as follows:*

> *"**The contrasting aerosols regimes observed at this site** and the very stable and low aerosol turbidity within the FT make it **an excellent** site for aerosol monitoring and calibration (Toledano et al., 2018; Cuevas et al., 2019b). Not in vain, Izaña Observatory, located in the FT, is considered one of the two absolute calibration sites in the world for both AERONET and GAW-PFR global networks (Toledano et al., 2018; Cuevas et al., 2019b)."*

**L136-7 Also, Solar Zenith Angle cut off for inversion product should be mentioned here, since it limits a large number of available data.**

*Authors:* *The authors agree with this comment. We have included in the text the following information:*

> *"...It should be noted that AERONET level 2.0 retrievals for SSA and **imaginary** refractive index are limited to AOD440nm > 0.4 **and solar zenith angles > 50º**, which limits strongly the amount of data available for aerosol characterization **(Sinyuk et al., 2020)...**"*

**L178-187 It is not consistent that different AE thresholds for dust laden conditions are set at the stations. In theory, AE should be independent of the station when saharan dust is dominating the aerosol mixture. This should be discussed in detailed. If there is some assumption related to the aerosol in the lower column, it should be thoroughly explained, since it affects the analysis of the next sections.**

*Authors:* *We agree with the referee that AE, as a qualitative indicator of the dominant aerosol particle size, is an intensive optical property that is independent on the station provided the same aerosols are present. This is not the case of Santa Cruz de Tenerife (coastal station, dominated permanently by marine aerosols and periodically by dust aerosols) and Izaña (high mountain station, characterized by background conditions and seasonally by Saharan dust). Therefore, we don't to have the same dominant aerosols in the two stations, and, as a consequence, thresholds for AE are not expected to be the same. These thresholds have been set and thoroughly studied in Barreto et al. (2022), so the authors consider that the explanation of these AE thresholds is not within the scope of the present paper.*

**L290. Since the criterion for FT is AOD500nm ≤ 0.10 , there should be no AERONET SSA retrieval at these cases. Please, explain if this SSA values are refering to FT or rephrase.**

*Authors:* *We agree with this referee that under clean FT conditions it is not possible to retrieve SSA. Following Sinyuk et al. (2020), SSA and imaginary refractive index are limited to AOD440nm > 0.4 and solar zenith angles > 50º. As it is stated in Sect. 3.2 of the manuscript, lines 203-204: "The reason for not including these parameters for background conditions is the high uncertainty in AERONET inversion products under low aerosol loading as reported by Sinyuk et al. (2020)". Line 290 corresponds to the section about dust-laden conditions and Fig. 9 (optical retrievals) is only for dust laden conditions. In line 287 we state: "The two stations in the FT, despite the low data availability, especially at TPO, exhibit a similar pattern in terms of the four variables. In the FT, the spectral dependence of g seems to be reduced...". We are referring to FT affected by the presence of SAL, and not to clean FT conditions. We propose the following changes in the manuscript to clarify this issue: "The two stations in the **dust-laden** FT, despite the low data availability, especially at TPO, exhibit a similar pattern in terms of the four variables. In **these stations**, the spectral dependence of g seems to be reduced".*

**L303 It is not very common to refer to some analysis that is not shown (at least in an appendix). Since it is important enough to be mentioned it in the manuscript, is should be presented.**

*Authors:* *Following the Referee's recommendations, we have added this analysis in the final manuscript as follows:*

> *"... This reduction is also consistent in terms of average AOD in dust-laden conditions (59.2%) and also considering a regression analysis of AOD difference between IZO and TPO against AOD at IZO, as the reference. In this analysis **(see Figure S1 in the supplementary material)**, a slope of 0.323 and a Pearson coefficient (R) of 0.75 are found..."*

[Figure]

*Figure S1.- Scatterplot of the difference between $AOD_{500nm}$ measured at IZO and PTO under dust-laden conditions versus $AOD_{500nm}$ measured at IZO. The fitting parameters are shown in the legend. R is the Pearson correlation and N is the number of data points.*

**3.3 It is valuable to present also the trends in frequencies of MABL, FT, Dust laden cases in those stations. Is there a change in the frequency of any of those layers, that is linked to the observed trends in optical properties?**

*Authors: We agree with the referee that it would be very interesting to include trends at the two stations and under the two regimes. However, we have not shown trends of the SSA and refractive index time series at SCO and IZO under dust conditions because we do not have enough data in Level 2.0 to do this study (SSA and refractive index at SCO: 293 days, SSA and refractive index at IZO: 45 days). We have included in the supplementary material the series of monthly mean values of total, fine and coarse-mode AOD at IZO and SCO because these are the only datasets suitable to perform trend analysis.*

**L332 The first break point of SO2 in 2009, is not convincing, it seems more like a 3 month parenthesis to the activities. In the 2009-2013 the higher SO2 values are recorded.  I suggest to just use two periods like in AOD fine mode fraction or discuss thoroughly the reasons for selecting 3 periods**

*Authors:*

*The first thing that the authors want to highlight is the robustness of the Lanzante Method (Lanzante, 1996) for determining break points. This is a known methodology widely used by the scientific community with more than 700 references. It involves the application of a non-parametric test, related to the Wilcoxon-Mann-Whitney test, followed by an adjustment step. After applying this technique, the 2005-2020 monthly mean SO2 series shows two break points. The first one occurs in February 2009 and the second break point took place in February 2013. Both break points are at 95% confidence level, and therefore the statistical significance of these results cannot be questioned. Furthermore, the change-point signal-to-noise ratios, which quantify the magnitude of relative importance of each discontinuity, are 2.52 and 2.73 (RDN), respectively, indicating that both breaks may be considered as the principal transition.*

*Regarding the comment of the referee about the possibility of a 3-month parenthesis as the possible cause of this break point in 2009, they authors want to emphasize that a detailed analysis of the causes of the changes in the trend of the SO2 series is completely outside the objective of this work. However, in order to give a response to the referee, we know that between 2007 and 2009 there was a 40% reduction in SO2 emissions from the refinery, from 2911 t to 1745 t (see Table 1 below) (CEPSA, 2011). This reduction in SO2 emissions was due to various measures implemented in the refinery, one of which was a reduction in the percentage of sulphur in the fuel oil used in the refinery from < 1% to < 0.7%, from 2008 onwards. Such changes in the refinery SO2 emissions can explain the switch to downtrend trend in SO2 concentration in these years. Subsequently, in 2013, the refinery ceased crude oil refining operations and this explains the second break point in 2013. The AOD fine mode trend includes other contributing emissions from the city of Santa Cruz de Tenerife, such as on-road traffic emissions, and so some of the changes observed in the SO2 time series are more masked in this component.*

**Table 1 Santa Cruz refinery SO2 emissions 2007-2011**

| Parameter | 2007 | 2008 | 2009 | 2010 | 2011 |
|---|---|---|---|---|---|
| SO2 emission (t) | 2911 | 2386 | 1745 | 1622 | 1002 |

CEPSA, 2011, Tenerife Refinery Environmental Declaration. (https://www.cepsa.com/stfls/CepsaCom/Coorp_Comp/Ficheros_corporativo/2012/AF%20TENERIFE%202011%20(baja).pdf)